# Morphology, Thermal and Mechanical Properties of Co-Continuous Porous Structure of PLA/PVA Blends by Phase Separation

**DOI:** 10.3390/polym12051083

**Published:** 2020-05-09

**Authors:** Natthapong Chuaponpat, Tsubasa Ueda, Akira Ishigami, Takashi Kurose, Hiroshi Ito

**Affiliations:** Graduate School of Organic Materials Science, Yamagata University, 4-3-16 Jonan, Yonezawa, Yamagata 992-8510, Japan; n.chuaponpat@yz.yamagata-u.ac.jp (N.C.); tra67198@st.yamagata-u.ac.jp (T.U.); akira.ishigami@yz.yamagata-u.ac.jp (A.I.); takashi.kurose@yz.yamagata-u.ac.jp (T.K.)

**Keywords:** polymer blend, phase separation, porous, scaffold, swelling, crystallization

## Abstract

Poly (lactic acid) (PLA) was blended with poly (vinyl alcohol) (PVA) in the composition of 70/30 (L7V3), 60/40 (L6V4), and 50/50 (L5V5) wt.%. L7V3 exhibits a sea–island morphology, while L6V4 and L5V5 show co-continuous phase morphologies. These polymers exhibited a solitary glass transition temperature, which obeyed the Fox equation. Thereafter, the blends were made porous by an etching process in hot water (35 °C) for 0–7 days, to remove PVA. The maximum etched PVA content of L7V3, L6V4, and L5V5 was 0.5%, 13.4%, and 36.1%, respectively; hence, L5V5 exhibited a co-continuous porous morphology with the porosity of 43.4%, the degree of swelling of 47.5%, and the pore size of 2 µm. The degree of crystallinity of PLA, exposed PLA, and L7V3 showed an insignificant change. L5V5, having the highest porosity, demonstrated the highest increase in the degree of crystallinity of approximately two times, because water induced the crystallization of PLA. The high porosity of L5V5 exhibited an excellent absorption property by increasing absorption energy more than two times, as obtained by micro indention. It had the maximum indentation depth more than 250 µm. Flexural and tensile properties considerably decreased with an increase in the porosity.

## 1. Introduction

Poly(lactic acid) (PLA) has attracted considerable attention as an alternative material for biomedical applications (e.g., surgical sutures, artificial skin, drug delivery materials, scaffolds, packaging, and tissue engineering) because PLA is renewable, processable, energy-saving, biodegradable, and biocompatible [1,2,3,4,5,6]. Tissue engineering relies on material properties and cell transportation to repair and regenerate bond defects [7,8]. Scaffold is used as a physical and biological support in tissue engineering, and it can be produced by PLA, poly (caprolactone) (PCL), poly(lactic-co-glycolic acid) (PLGA), poly (vinyl alcohol) (PVA), poly (butylene succinate) (PBS), and poly (hydrobutyrate) (PHB) [9,10,11,12,13,14,15,16,17]. Mathieu et al. [9] produced composite scaffold of PLA and ceramic powder for bone tissue engineering by using supercritical carbon dioxide as a physical foaming agent. Biocompatibility with human primary osteoblasts was evaluated by cell colonization and expression of ribosomal 18S gene. Cells proliferated on all the composite scaffolds, so it was pointed that the composite scaffolds were biocompatible. Gregor et al. [10] fabricated PLA scaffolds by a fused deposition modeling, using a 3D printer; two types of scaffold structures were prepared, and their porosities were 30% and 50%. They were sufficient proliferation of osteosarcoma cells even though a recommended porosity of scaffold for bone tissue was 90%. Pan and Ding [13] reviewed scaffold of PLGA for tissue engineering and regenerative medicine and proposed four fabrication techniques at moderate and room temperature. Zhou et al. [15] studied PVA scaffold by using quenching and a freeze-drying technique. The scaffold morphology obtained from 18 wt.% PVA solution was a unidirectional honeycomb-like structure with a porosity of 71%. Then they tested the scaffold’s drug-release efficiency, and it had the release efficiency of 54.5%. When it was added to poly (ethylene glycol) (PEG), the porosity increased from 71% to 81%, and the release efficiency was improved, by increasing from 54.5% to 89.3%. Thus, PVA/PEG scaffold can be utilized for tissue engineering. For tissue-engineering applications, scaffold properties are essential and need to exhibit the required properties of biocompatibility, pore size, pore size distribution, pore interconnection, mechanical properties, and biodegradability [18]. The pore-size parameter is very important, and it is suitable for new tissue growth and reorganization [19]. A scaffold provides support during tissue reorganization. Thus, mechanical properties need to be optimized. These properties were summarized by Kramschuster and Turng [20].

Scaffolds can be obtained by textile processes, porogen leaching, and phase separation. Phase-separation processes are used to obtain scaffolds by the thermal instability of polymer solution or polymer blends. These methods use an organic solvent, which is a critical parameter because, if the solvent is not fully removed, it may affect tissue reorganization [21,22]. Hence, water-soluble polymer blends are utilized, such as poly (ethylene oxide) (PEO), PVA, and PEG [11,20,23,24,25,26,27,28]. Huang et al. [11] prepared interconnected porous poly (ε-caprolactone) (PCL)/PEO by incorporating the foaming process and PEO leaching in deionized water. The porosity can reach 89.5% at 50% of PCL. Even though the compression modulus depressed from 68.2 to 46.7 MPa, it met the compressive modulus requirement. Kramschuster and Turng [20] have also used these techniques to produce a PLA/PVA/NaCl composited scaffold. The developed scaffold achieved the maximum porosity of 75%. Aramwit et al. [23] produced a Sericin/PVA/glycerin scaffold by using the salt (NaCl) leaching method, which is a solvent-free, low-energy-consumption, short-leaching-time, and low-cost method compared to the freeze-drying method. Thadavirul et al. [25] incorporated NaCl and PEG as water-soluble porogens, to create a PCL scaffold. The cooperation of NaCl and PEG let the scaffold reach the porosity of more than 90%. Thadavirul et al. also pointed out that PEG improved interconnectivity and cell support. Sun et al. [27] proposed a novel methodology to provide hierarchically porous of polylactide monoliths. Poly (L-lactide acid) (PLLA) was blended with poly (D-lactide) (PDLA), to obtain stereocomplex crystallite, which induced the hierarchical morphologies and blended with PEO as porogen. Additional PDLA let stereocomplex crystalline increase from 13.1% to 61.2%, resulting in an increase of specific surface area and pore volume by more than two times after removing PEO. Moreover, increasing PEO content at 70/30 wt.% of PLLA/PDLA enhances porous morphologies by raising the specific surface area and pore volume by more than 80 and 300 times, respectively. Zhang et al. [28] also developed a methodology to obtain porous high-density poly (ethylene) bundles by blending and leaching PEO. An extruded HDPE/PEO was extracted in deionized water, to remove PEO, and peeled off to obtain a porous HDPE bundle with a villus-like structure. The porous bundle improved in hydrophobicity by increasing the water contact angle up to 139°, and super lipophilicity improved by decreasing the cyclohexane contact angle from 14° to 0°. It performed high oil absorption and pumping-oil-recovery ability. Therefore, it had highly potential application in oil–water separation. PVA is widely used in many industrial applications, owing to its biocompatibility, biodegradability, thermal stability, nontoxicity, and excellent mechanical properties [29]. Moreover, PVA can be etched by water because it contains a hydroxyl group so that PVA in the phase-separated sample can be removed by the etching process in water, which provides a solvent-free method to obtain the scaffold of the PLA/PVA blend.

Depending on scaffold properties, blended morphology is essential to specify the scaffold application because it affects porosity and mechanical properties. The immiscible polymer blend has four types of morphology, i.e., matrix-dispersed particle (sea–island), matrix-fiber, lamellar, and co-continuous [30]. Different morphologies affect mechanical properties; thus, polymer blends can produce desired mechanical properties. Willemse et al. [31] reviewed and summarized the following characteristics. Specifically, the sea–island morphology improves impact properties; the lamellar morphology enhances barrier properties, and the mechanical properties of the co-continuous morphology are between those of a blended polymer. Willemse et al. concluded that phase inversion from sea–island to co-continuous morphologies depends on interfacial tension, viscosity, volume fraction, shear rate, and phase dimension. A fully interconnecting porous polymer can be improved and enhanced by porogen particle leaching, annealing and foaming processes, and thermally induced phase separation [32,33,34]. Gao et al. [35] investigated the effects of three parameters (blend ratio, heat-treatment temperature, and time) on the phase structure of poly (L-lactide acid) (PLLA)/poly (propylene carbonate) (PPC) blends by an optical microscope. The most significant factors that affect the phase structure are the heat-treatment temperature and time. A bi-continuous phase structure can be observed at 60/40 wt.% (PLLA/PPC) at the heat-treatment temperature of 200 °C for 5 min.

As mentioned above, PLA and PVA can be utilized in tissue engineering application due to biocompatibility; hence, a PLA/PVA scaffold obtained in the solvent-free method should be investigated, to evaluate its mechanism and property change. In this study, PLA and PVA blends with varying PVA compositions were prepared, and the co-continuous porous structure was governed by an etching process in hot water, to remove PVA. PVA can be dissolved in water; hence, water diffused and penetrated the structure. Blending and water penetration affect morphologies and mechanical properties. Hence, this study aimed to investigate the morphology and the change in physical, thermal and mechanical properties of PLA/PVA blends owing to the etching process in hot water. Injected pure PLA was also exposed to hot water, for the comparison of properties. Pure PLA and PVA were prepared, and their thermal and mechanical properties were characterized and compared to those of their blends.

## 2. Materials and Methods

### 2.1. Materials

PLA (grade TE-2000) with a melting point of 170 °C was purchased from UNITIKA Ltd., Osaka, Japan. PVA (grade MOWIFLEX^TM^ C17) was purchased from Kuraray Co., Ltd. Tokyo, Japan.

### 2.2. Sample Preparation

Neat PLA and PVA were dried and preheated at 80 °C for 5 h and at 60 °C for 6 h, respectively. They were dry-mixed at different compositions, which are presented in Table 1, and blended by using a twin-screw extruder (KZW15TW-30MG-NH (-700) from TECHNOVEL CORPORATION, Osaka, Japan) at the following conditions: feed to die temperature of 40–170 °C, feeding rotation speed of 12 rpm, screw rotation speed of 500 rpm, and naturally cooling temperature of 20 °C (room temperature). Then, extruded samples were cut into a pellet shape. These samples were processed by injection molding (EP5 model from NISSEI PLASTIC INDUSTRIAL CO., LTD., Nagano, Japan), to obtain dumbbell and sheet shapes at the following conditions: feed to die temperature of 40–180 °C, mold temperature of 40 °C, injection pressure of 45 MPa, holding pressure of 40 MPa, and injection speed of 10 mm/s. Dimensional samples are illustrated in Figure 1.

### 2.3. Phase Separation

The samples were weighed ($m_{0}$), immersed in distilled water, and stirred at 35 °C for 0, 1, 3, 5, and 7 days, to etch PVA; distilled water was replaced every 6 h. Etched PVA samples were dried in a vacuum dryer at 40 °C for 24 h, to remove moisture; thereafter, they were weighted (m) to calculate the etched PVA content (ϖ) by Equation (1).
(1)$$\varpi =\left(\frac{m_{0}-m}{m_{0}}\right)\times 100$$

### 2.4. Characterization

#### 2.4.1. Rheology

Neat PLA and PVA were preheated, and the melt viscosity was measurement by a capillary rheometer (Capilograph 1D, Toyo Seiki Seisaku-Sho Ltd., Tokyo, Japan) at 180 °C, with the orifice diameter of 1 mm and the capillary length of 10 mm.

#### 2.4.2. Morphology

The samples were cut at cryogenic conditions (by employing liquid nitrogen), to observe the morphology of the cross-sectional area of the samples. Owing to the low conductivity of PLA/PVA, the samples were coated by Ar plasma from an ion bombarder (PIB-10, Vacuum Device Co., Ltd., Ibaraki, Japan). Then, the samples were observed by using scanning electron microscopy (SEM) (TM3030Plus, Hitachi High-Tech Corporation, Tokyo, Japan).

#### 2.4.3. Flexural and Tensile Strength

The sheet and dumbbell samples were prepared, and their flexural and tensile strengths were characterized, respectively. Flexural and tensile strengths were evaluated by STROGRAPH VGS1-E (Toyo Seiki Seisaku-Sho Ltd., Tokyo, Japan) with a crosshead speed of 5 and 1 mm/min, respectively, and a 1-kN load cell. The span length of flexural testing was 30 mm.

#### 2.4.4. Indention

The sheet samples were subjected to indention, using a micro indenter (Micro Compression Tester (MCT), SHIMADZU CORPORATION, Kyoto, Japan) with a spherical tip (50 µm radius), at the applied force of 5000 mN.

#### 2.4.5. Density

The density was measured by a pycnometer, which relied on the Archimedes law and an electrical balance (AD-1653 model, A&D Company, Ltd., Tokyo, Japan) readable to 1 mg.

#### 2.4.6. Differential Scanning Calorimetry (DSC)

The 4.0–5.0 mg samples were prepared and characterized by DSC Q200 (TA Instruments Japan Inc., Tokyo, Japan), to investigate thermal properties. Scanning temperature was in the range of 40–200 °C, at the constant ramping rate of 10 °C/min; nitrogen gas was used as an inert atmosphere to prevent oxidation. Two heating–cooling cycles were performed to analyze the thermal properties of the samples.

The degree of crystallinity of PLA (*X_c_*) was evaluated and calculated by Equation (2). There is a correlation between the enthalpy of melting ($\Delta H_{m}$), enthalpy of cold crystallization ($\Delta H_{cc}$), weight fraction of PLA (ω), and enthalpy of melting of 100% of crystalline PLA ($\Delta H_{m}^{0}$) (93 J/g) [36]. It is correct to use the sum of $\Delta H_{m}$ and $\Delta H_{cc}$, because the contributing part also contributes to $\Delta H_{m}$.
(2)$$X_{c}=\frac{\Delta H_{m}+\Delta H_{cc}}{\omega \times \Delta H_{m}^{0}}\times 100$$

## 3. Results and Discussion

### 3.1. Rheology and Morphology

PLA/PVA samples extruded from the twin-screw extruder were used to observe the morphology, and their morphology is shown in Figure 2a–c. The extruded samples exhibited a sea–island morphology, which classifies the samples as immiscible polymer blends [37]; micro voids were observed at all cross-sectional areas. It was suggested that the PVA phase obtained in extruded L7V3 and L6V4 was covered by the PLA phase, owing to low concentration and high viscosity, as shown in Figure 3. The viscosity and experimental shear rates of PVA were higher than those of PLA. Therefore, the PLA phase was more deformed than the PVA phase during melt blending by the twin-screw extruder. An increase in the amount of PVA up to 50 wt.% resulted in the volume fraction being close to unity, and extruded L5V5 appeared to have fewer micro voids (Figure 2c). The extruded samples were subjected to the injection process, to produce sheet and dumbbell samples; then, the sheet samples were used to investigate the effect of processing on the morphology; the results are shown in Figure 2d,e. The injected samples produced by the application of a higher shear rate resulted in a considerable decrease in the micro voids, especially in L5V5 (Figure 2e). The addition of PVA and further processing resulted in the specific morphology of PLA/PVA blends.

To obtain a co-continuous porous morphology, the PLA/PVA blends were immersed in hot water (35 °C) to etch PVA, because it has a hydroxyl group in the molecular structure; thus, PVA can be dissolved and removed by water. Moreover, mass transportation and swelling by water uptake are responsible for this process because water molecules are smaller than polymer molecules [38]. These mechanisms have been described by Alfrey et al. [39] and have been also classified into case I (Fickian diffusion) and case II, respectively. Moreover, combination mechanism is defined as anomalous diffusion. To observe swelling, the thickness of sheet samples was measured in a normal direction (ND) ($D_{ND}$) near the edge (E) and core (C), to evaluate the swelling degree in ND ($S_{ND}$) by Equation (3). A relationship between the swelling and etching time of the blends is shown in Figure 4. Both swelling degrees at the edge and core of L7V3 were less than 1% because water uptake could not penetrate deeply, owing to the co-continuous PLA phase in the sea–island morphology, which acted as a barrier to prevent dissolution and etching of PVA. When PVA content was increased up to 40 wt.%, the swelling degrees at the edge and core at 1 day were less than 2%, which was due to etching on the contact surface between the sample and water. At the etching time of 3 days, the swelling degree at the edge was higher than that at the core because water penetrated the edge by three paths (i.e., top, bottom, and side), while it penetrated the core by only two paths (top and bottom). Thus, more PVA can be removed at the edge than at the core, and the sample exhibited higher swelling at the edge. Then, the degree of swelling at the core continuously increased, while that at the edge leveled off after etching for 5 days. The swelling degree at the core for the etching time of 7 days was slightly larger than the swelling degree at the edge. For L5V5, the swelling degree at the edge considerably increased up to 20%, then slightly increased and leveled off at the etching time of more than 3 days; the swelling degree at the core continuously increased and tended to level off at the etching time of 5 days. A considerable increase in the degree of swelling for 3–5 days of etching was governed by the sudden change in chemical potential and stress [40,41], owing to the merging of two water fronts. Therefore, porous materials obtained in the water-etching process should be concerned due to the swelling effect.
(3)$$S_{ND}=\left(\frac{D_{ND}-1}{1}\right)\times 100$$

Figure 5 shows an etched PVA content (ϖ) from the blends, along with etching times. The etched PVA content of L6V4 and L5V5 considerably increased; however, the etched PVA content of L7V3 changed insignificantly with an increase in the etching time. The maximum etched PVA content of L7V3, L6V4, and L5V5 was 0.5%, 13%, and 36%, respectively. In addition, a sample becomes fully interconnecting porous if the etched PVA content has an equal weight fraction. These results show that the etched PVA content increased with an increase in the PVA content and etching time. The etched PVA content produces co-continuous and interconnecting porous morphologies because the co-continuous PVA phase lets water permeate and penetrate deeply. There was 14 wt.% PVA remaining in L5V5; hence, porogen as NaCl can be incorporated in improving the etched PVA content, as was mentioned in the Introduction section. L5V5 maintained high PVA content, which resulted in co-continuous porous morphology. Porous morphology and swelling resulted in voids inside the structure; hence, porosity was measured to evaluate the porous structure. Porosity is defined as the ratio of void volume and total volume and is expressed in Equation (4), which is a relationship between before ($\rho_{0}$) and after (ρ) etching density of the sample. Figure 6 shows the porosity of PLA/PVA blends. The results show that porosity was proportional to the etched PVA content. Porosity continuously increased along with the etched PVA content and was not related to the amount of blended PVA. Moreover, an increase in the porosity from 32% to 42% for L5V5 was an extreme shift, owing to the merging of two water fronts and a sudden change in chemical potential and stress that resulted in more swelling. Large volume appearance implied that the density reduced, and porosity increased, respectively. The PVA etching process had the samples expand; thus, it should be considered.
(4)$$\phi =\left(1-\frac{\rho }{\rho_{0}}\right)\times 100$$

The cross-sectional area of the samples was evaluated by SEM, to observe morphology at the transverse (TD) and flow (FD) directions, and the observed locations are shown in Figure 7. TD and FD morphologies were similar; thus, TD SEM micrographs at the core were selected for discussion, and SEM micrographs at the (I) and (II) locations of TD are shown in Figure 8. It was difficult to observe co-continuous porous L7V3 during the entire experiment because PVA was not etched at the experimental conditions. Co-continuous porous L6V4 could be observed at the etching time of 7 days, and it appeared at the (II) location. Its region is indicated by a white double-arrow in the figure, and its depth from the surface was approximately 150 µm. L5V5 at the (II) location was co-continuous porous at the etching time of 1 day with a 100 µm depth, whereas the co-continuous porous L5V5 at the (III) location shown in Figure 9 was clearly recognized. The co-continuous porous and non-porous border is divided by a dashed line, and the approximate depth was greater than 200 µm. These results indicate that water penetrated the (III) location by three paths (i.e., top, bottom, and side); however, water penetrated the (II) location by two paths (i.e., top and bottom). Thereafter, the depth could not be classified and evaluated, because two water fronts merged; thus, L5V5 became co-continuous porous. However, a sudden change in the chemical properties made the sample more expanded at the merging zone. Cracking morphology appeared at the merging zone, and it was caused by cutting during cryogenic cracking. Swelling and morphology results showed that water uptake penetrated through the blends by the co-continuous PVA phase, allowing the sample to expand along the penetration path and resulting in porous morphology. Water uptake penetration is shown in Figure 10.

The co-continuous porous morphology of L5V5 at high magnification for the etching time of 7 days is shown in Figure 11. An observed black hole was interconnecting porous and was represented the etched PVA. It could be pointed out that PVA formed an island phase and PLA formed a matrix phase. There was sufficient time for PVA etching and water penetration; hence, the samples became co-continuous porous. Pores at the (II) location for both TD and FD had elliptical shapes, owing to the deformation of the PVA phase near the surface. These results were selected to evaluate the pore size by employing the free software (Image-J), and the pore size distribution curve is shown in Figure 12. (II)-TD, (II)-FD, and (I)-FD curves shifted and deviated from the (I)-TD curve, and their mean values were close to 2 µm. The (II)-FD mean was slightly larger than the (II)-TD mean, and the weighted average pore sizes of (II)-FD, (I)-FD, (II)-TD, and (I)-TD were 2.5, 2.3, 2.0, and 1.5 µm, respectively. In addition, the pore size at the (II) location was larger and more distributed than the pore size at the (I) location, because dispersed PVA phase was deformed due to shear and temperature gradients in a melt during the injection mold process [42].

### 3.2. Thermal Properties

Second heating scan thermograms of pure injected PVA, PLA, and their blends are shown in Figure 13, and thermal data obtained from these thermograms are presented in Table 2. Injected PVA exhibited the glass transition temperature (T_g,PVA_) of 57.0 °C and a broadly endothermic curve with the melting temperature (T_m,PVA_) of 150.3 °C. The thermal properties of injected PLA were as follows: T_g,PLA_ of 62.2 °C, cold crystalline temperature (T_cc_) of 108.8 °C, and T_m,PLA_ of 169.0 °C. The blends exhibited solitary T_g_, and it was intermediate between T_g,PVA_ and T_g,PLA_. In a miscible blend system, depending on the T_g_ and composition of the blend material, it appears as a single Tg according to Fox’s equation [43]. On the other hand, in immiscible blend systems that form co-continuous or sea–island structures, the Tg of the blend material appears independently. We plotted the experimentally determined Tg and the calculated T_g_ by the Fox equation in Figure 14. The T_g_ of these blends is slightly different from the Fox equation, which may indicate that they are partially miscible in the amorphous phase. When the heating temperature increased, cold crystallization was observed in DSC thermograms. The T_cc_ of blends slightly decreased from approximately 108 to 102.4 °C when the blended PVA content increased from 30 to 50 wt.%. The melting temperature of the blends clearly exhibited the presence of two peaks corresponding to T_m,PVA_ and T_m,PLA_. To observe an endothermic peak at T_m,PLA_, non-smooth curves or multiple endothermic peaks of PLA were monitored; the process involved crystalline phase transition from α’ to α phases [44]. Zhang et al. investigated and characterized the phase transition of poly (L-lactice) (PLLA) by using WAXD and DSC. They reported that phase transition occurred when the crystallization temperature (T_c_) of PLLA was in the range of 100–120 °C. The T_c_ of L7V3, L6V4, and L5V5 were 115.4, 115.1, and 113.6 °C; therefore, phase transition could be observed at the endothermic peak of PLA.

The degree of crystallinity (X_c_) of the sheet sample was characterized at the edge and core, to observe the effect of etched PVA on the crystalline content. The injected sheet of pure PLA samples was subjected to the water-etching process at 35 °C for 7 days and denoted as PLA7. PLA7 at the edge (PLA7E) and core (PLA7C) contained the PLA crystalline content of 12.4% and 12.3%, respectively, which did not considerably change by injected PLA (12.6%). Water could not plasticize and induce crystallization in both PLA7E and PLA7C. Similar results have been published by Iñiguez-Franco et al. [45]. They proposed that the crystallization of PLA was induced by pure water, and the crystallinity increased by 3%, while the crystallinity of PLA exposed to ethanol solutions considerably increased from 2.3% to more than 40% at the exposure time of 180 days. After the etching process, water uptake made the samples swell; thus, there were physical and thermal changes in the structure. The degree of swelling was evaluated, to observe physical change, and the degree of crystallinity at the etching time of 0, 1, and 7 days was calculated, to investigate thermal change. According to Equation (2), the weight fraction of the etched blends was calculated from the remaining PLA content after the etching process. Their relationship is shown in Figure 15. The crystalline content of PLA clearly increased with an increase in the swelling by water uptake. The degree of crystallinity of L7V3 remained constant after the etching process; the degree of swelling was determined to be less than 2%. However, more PVA in L6V4 and L5V5 could be removed during the etching process, resulting in high swelling and crystalline content. The degree of crystallinity of L6V4E and L6V4C after 1 day of etching increased from 9% to 12% and 11%, respectively; then, it slightly increased to 13% and 12%, respectively, after 7 days of etching, with an increase in the degree of swelling up to 22%. Moreover, the degree of crystallinity of L5V5E and L5V5C considerably increased after the first day of etching, from 9% to 18% and 16%, respectively. Thereafter, it slightly increased and exhibited the PLA crystalline content of 19% and 16%, respectively, after 7 days of etching. These results can be summarized into three topics: (1) the PLA crystalline structure could not be plasticized by water, but it could be induced by the combination of blending and etching processes, because the blends were miscible in the amorphous phase; hence, water uptake could penetrate through a continuous PVA phase; (2) the PLA crystalline structure was induced, and its content increased at the beginning of etching; with an increase in the etching time, it was insignificantly affected by water uptake; (3) the PLA crystalline structure at the edge was more plasticized and induced by water uptake than the core, owing to transport limitation at the core.

### 3.3. Mechanical Properties

A change in the morphology of material makes the material exhibit different mechanical properties. The blends were subjected to microhardness, bending, and tensile tests, and the relationship between microstructure change (porosity and degree of swelling) and mechanical properties were characterized. An example of the micro-indenter testing curve of the blends at the etching time of 7 days is shown in Figure 16. Non-porous L7V3 required a shallow depth of approximately 50 µm to reach the applied load of 5000 mN. The depth of L6V4 at the edge and core was less than 150 and 100 µm, while co-continuous porous L5V5 had the depth of more than 250 µm. The co-continuous porous morphology was categorized as a soft material, which could absorb more energy than a rigid material. Therefore, the integration of the applied load and indentation depth curve represents absorption energy, and a relationship between the absorption energy and the degree of swelling is shown in Figure 17. The absorption energy of injected PLA and PVA was almost 0.10 mJ for both the edge and the core. There was low swelling in L7V3; therefore, the absorption energy of L7V3 was in the narrow range of 0.09–0.15 mJ. The absorption energy of L6V4 at the core slightly increased from 0.1 to 0.18 mJ; then, it remained at this value, because the depth of an indenter tip reached approximately 100 µm when it approached through the non-porous morphology. The absorption energy of L6V4 at the edge was more than that at the core, and the depth was up to 150 µm because L6V4 at the edge has a more porous morphology than that at the core. L5V5 exhibited highly co-continuous porous morphology with the degree of swelling of more than 30% and with the maximum absorption energy of 0.46 mJ. However, the co-continuous porous morphology of L5V5 resulted in similar absorption energies at the core and edge regions. These results show that the highly porous morphology had higher absorption energy. The edge region was more porous than the core region; thus, it had higher absorption energy.

Flexural and tensile tests were conducted to evaluate the mechanical properties of injected PLA, PVA, and their blends. The mechanical properties of injected samples with and without PVA etching were summarized and are presented in Table 3. Injected PLA was also etched for 7 days, at the conditions described in Section 2.3, to compare with the etched samples; this sample was denoted as PLA7. Injected PVA exhibited outstanding flexural and tensile strengths compared to injected PLA and their blends. The mechanical properties of the blends were reduced and are related to the strengths of injected PLA, owing to the PLA rich phase. The strengths of the blends considerably decreased, along with an increase in the PVA content, which suggested that the degree of phase separation reached the maximum value [46]. They investigated the miscibility of PVA and PLLA with a composition of 100/0, 90/10, 70/30, 50/50, 30/70, 10/90, and 0/100 (PLLA/PVA). They determined that the tensile strength of PLLA/PVA blends decreased with an increase in the PVA content; it reached a minimum at the composition of 50/50; then, it increased until the composition of PVA of 100 wt.%. After the etching process, the mechanical properties of the blends considerably decreased with an increase in the porosity. The relationship between the mechanical properties and the porosity is shown in Table 4. The mechanical properties of PLA7 were slightly decreased by less than 5% compared with those of injected PLA, owing to hydrolytic degradation [43]. Thus, a decrease in the mechanical properties of the blends was mainly governed by a change in the morphology. The mechanical properties of L6V4 were better than those of L5V5 because the L6V4 morphology was composed of two phases, solid and co-continuous porous phases; the solid phase was covered by the co-continuous porous phase. Hence, the mechanical properties of L6V4 were between those of solid and porous morphologies. L5V5 with the porosity of more than 10% exhibited a considerable decrease in the tensile strength and elongation at break because it contained thin walls and a highly porous morphology. Thus, the sample required low tension force to break. Figure 18 shows a ripped structure of L5V5 with the porosity of 43.4%.

## 4. Conclusions

Miscible PLA/PVA blends with a composition of 70/30 (L7V3), 60/40 (L6V4), and 50/50 (L5V5) wt.% were prepared by employing the twin-screw extruder and injection molding. A combination of postprocessing by injection molding and an increase in the PVA content improved blend morphology by phase conversion from sea–island to co-continuous morphology. A solitary glass transition temperature was observed that was intermediate between those of pure PLA and PVA. The blends were etched to remove PVA from hot water. L6V4 and L5V5 samples swelled more at the edge than at the core at the beginning of etching; then, the core of the samples swelled more than the edge. The degree of swelling was indicative of the water uptake penetration. The degree of swelling of L5V5 at the core reached the maximum value of 47.5%; thus, L5V5 exhibited the maximum porosity of 43.4% and the etched PVA content of 36.1%. L6V4 had the maximum degree of swelling of 22%, porosity of 25%, and etched PVA content of 13.4%. L7V3 had the maximum degree of swelling of 2%, porosity of 2%, and etched PVA content of 0.5%. L5V5 preferred to have a co-continuous PVA phase; therefore, L5V5 exhibited a co-continuous porous morphology with the weighted average pore size of approximately 2.5 µm. The majority of water uptake was done by PVA during the etching process; however, water also induced the PLA crystalline structure in the amorphous phase. L5V5 obtained the maximum etched PVA content, which was strongly affected by the solvent (water)-induced crystallization; thus, the degree of crystallinity increased from 9% to 18.6%. The advantage of the porous morphology was high absorption energy, which was demonstrated by the micro-indenter testing curve. L5V5 exhibited the maximum absorption energy of 0.46 mJ, while the maximum absorption energy of L6V4 was 0.28 mJ. However, the flexural and tensile properties of porous morphology considerably decreased with an increase in the porosity. The L6V4 structure exhibited heterogeneous morphology and contained porous and solid regions; solid regions were enclosed by porous regions; thus, mechanical properties were intermediate between those of porous and solid regions. These investigations found that the water-etching process made the samples swell at the merging water fronts; thus, the swelling effect due to the water etching process should be considered.

## Figures and Tables

**Figure 1 polymers-12-01083-f001:**
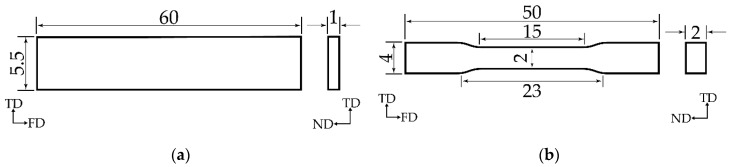
Dimensional samples (unit: mm): (**a**) sheet sample; (**b**) dumbbell sample.

**Figure 2 polymers-12-01083-f002:**
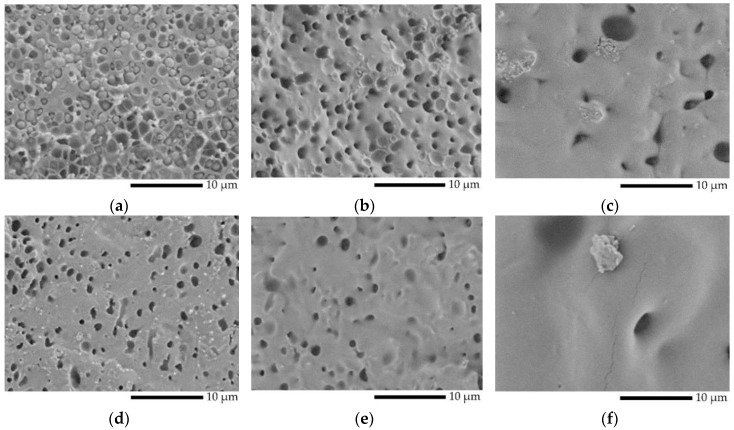
SEM micrographs of PLA/PVA blends at magnification of 5000x: (**a**) extruded L7V3; (**b**) extruded L6V4; (**c**) extruded L5V5; (**d**) injected L7V3; (**e**) injected L6V4; and (**f**) injected L5V5.

**Figure 3 polymers-12-01083-f003:**
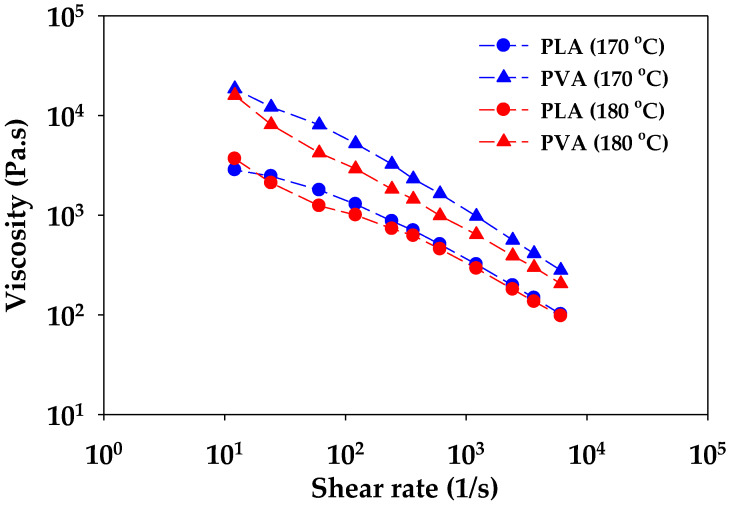
Melt viscosity of pure PLA and PVA at 170 and 180 °C.

**Figure 4 polymers-12-01083-f004:**
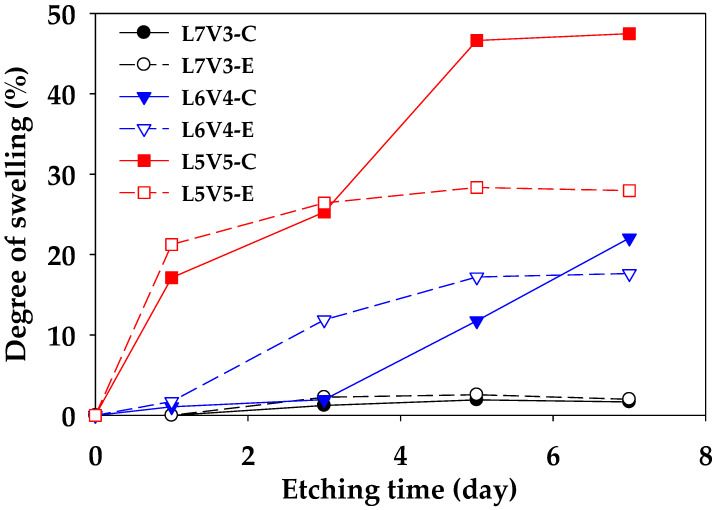
Degree of swelling in ND of PLA/PVA blends, along with etching time.

**Figure 5 polymers-12-01083-f005:**
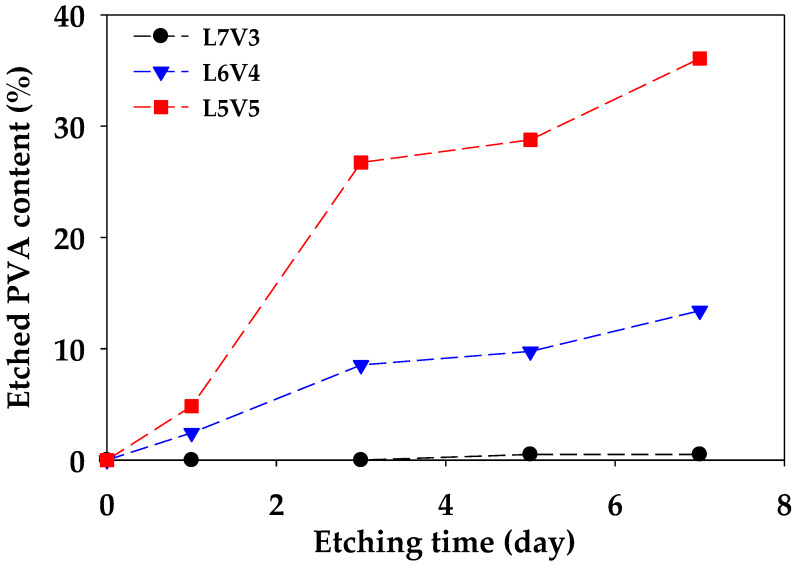
The etched PVA content of PLA/PVA blends, along with etching times.

**Figure 6 polymers-12-01083-f006:**
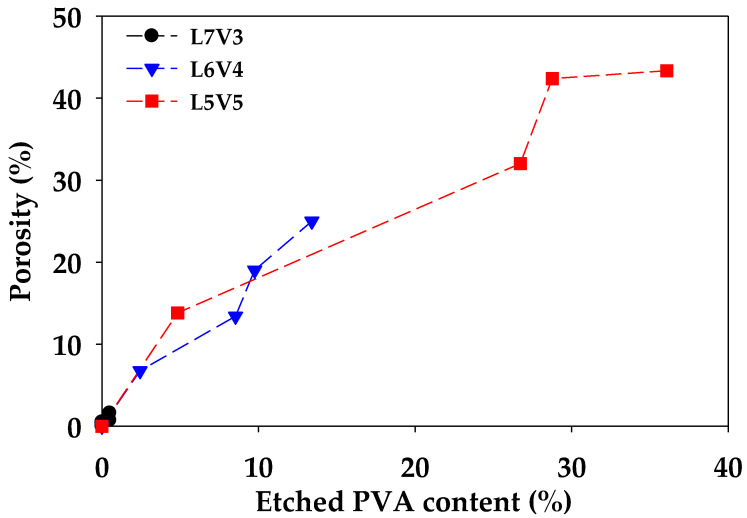
Porosity as a function of the etched PVA content of the blends.

**Figure 7 polymers-12-01083-f007:**
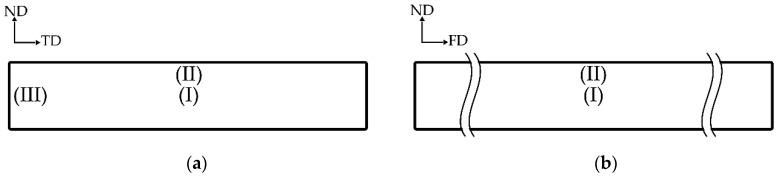
Observed SEM morphology location of sheet sample: (**a**) transverse direction (TD) and (**b**) flow direction (FD).

**Figure 8 polymers-12-01083-f008:**
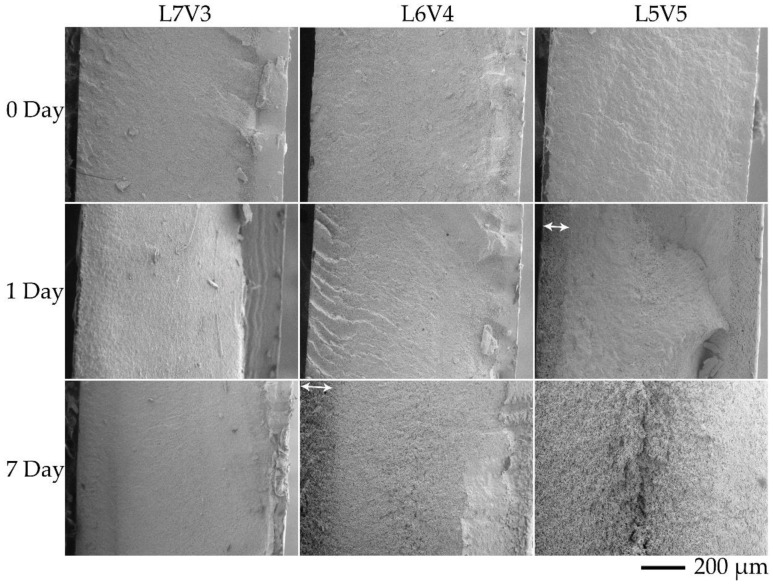
SEM micrographs of the blends in transverse direction (TD) at (I) and (II) locations.

**Figure 9 polymers-12-01083-f009:**
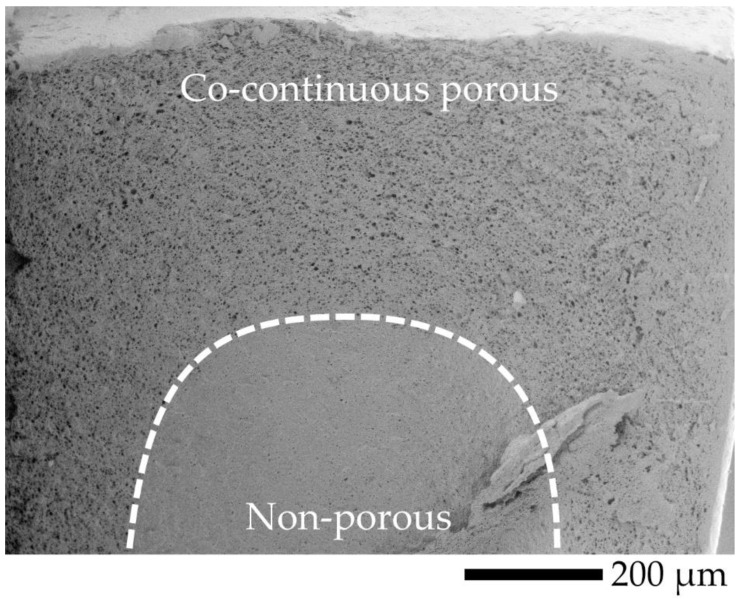
SEM micrograph observed at (III) location of L5V5 in TD at etching time of 1 day.

**Figure 10 polymers-12-01083-f010:**
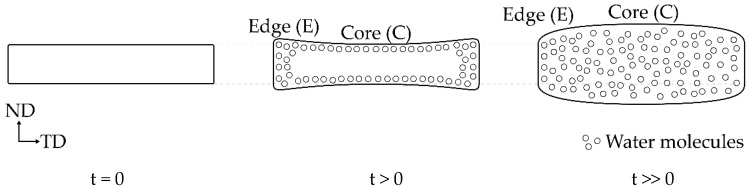
Swelling of a sheet sample during PVA etching time (t) in hot water.

**Figure 11 polymers-12-01083-f011:**
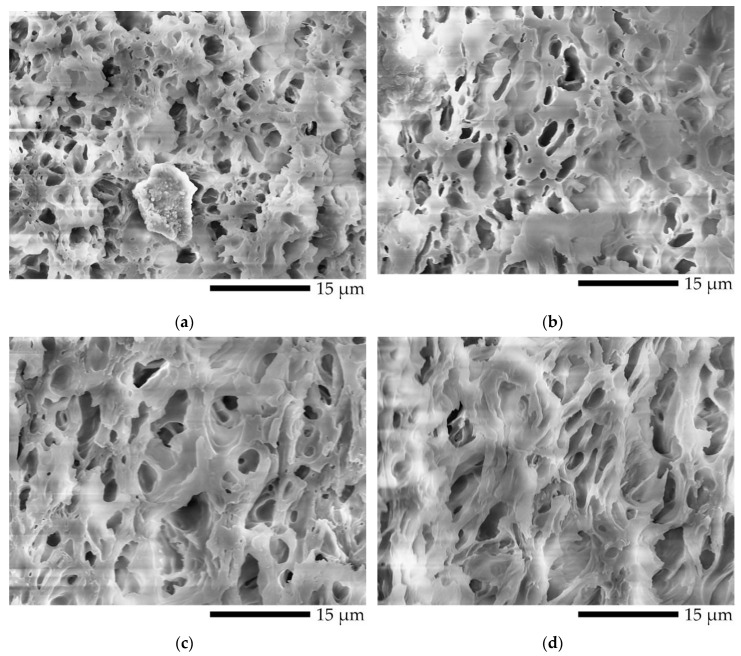
Co-continuous porous of L5V5 at different location at etching time of 7 days: (**a**) (I) location in TD; (**b**) (II) location in TD; (**c**) (I) location in FD; and (**d**) (II) location in FD.

**Figure 12 polymers-12-01083-f012:**
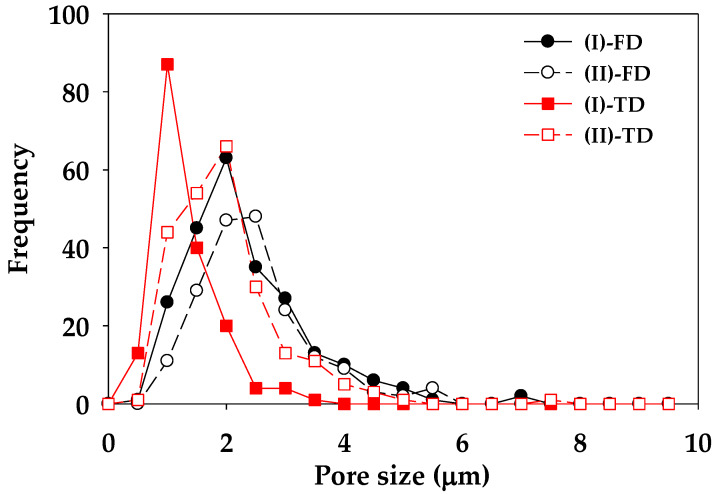
Pore size distribution curves of L5V5 at etching time of 7 days.

**Figure 13 polymers-12-01083-f013:**
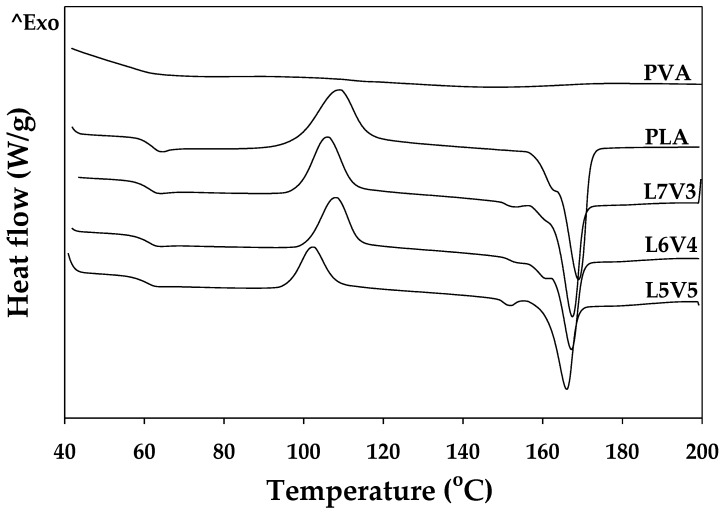
Second heating scan DSC thermograms of injected samples.

**Figure 14 polymers-12-01083-f014:**
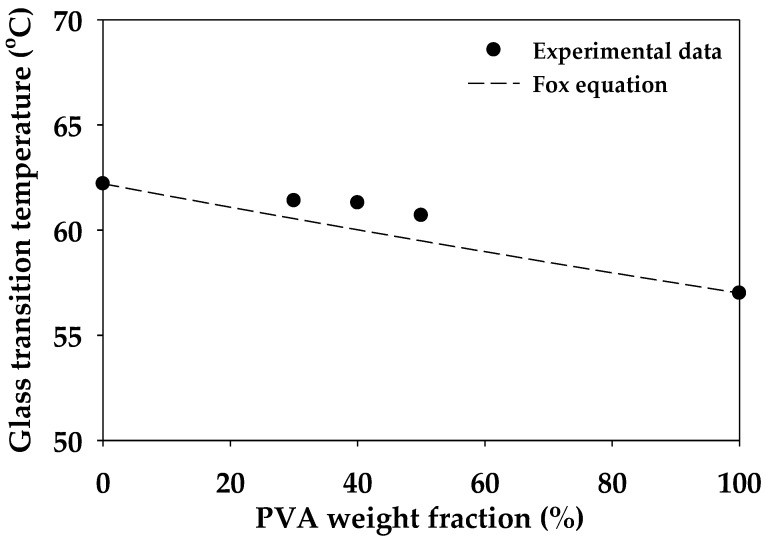
Relation between PVA weight fraction and T_g_ of PLA, PVA, and their blends.

**Figure 15 polymers-12-01083-f015:**
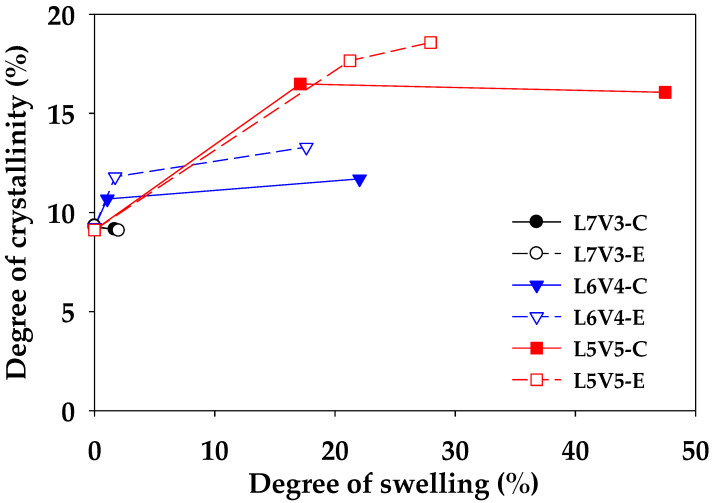
Relationship between the degree of crystallinity and degree of swelling.

**Figure 16 polymers-12-01083-f016:**
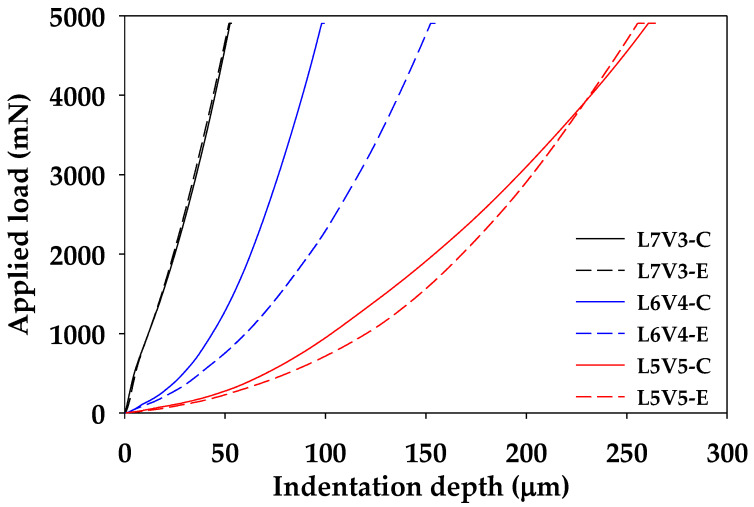
Micro-indenter testing curve of the blends at etching time of 7 days.

**Figure 17 polymers-12-01083-f017:**
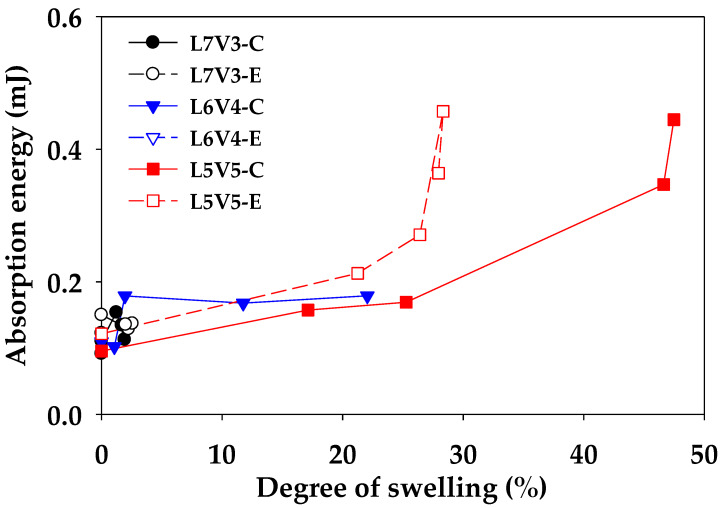
Relationship between the absorption energy and the degree of swelling.

**Figure 18 polymers-12-01083-f018:**
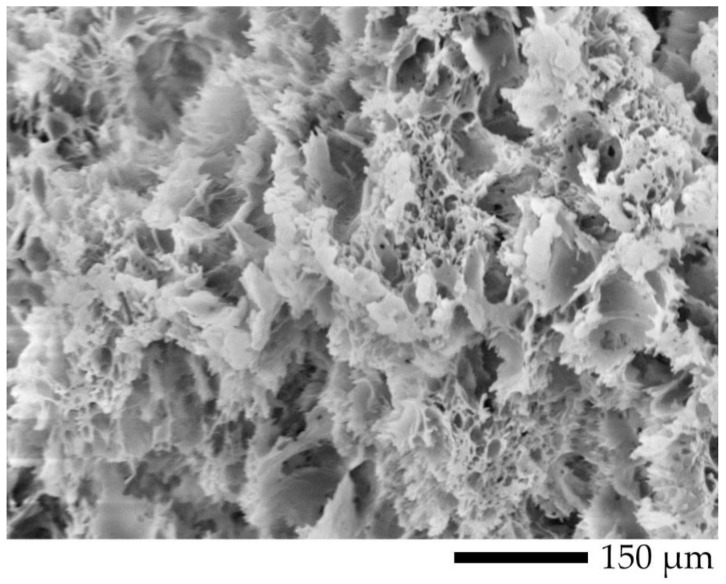
Tensile cracking morphology of L5V5 at the porosity of 43.4%.

**Table 1 polymers-12-01083-t001:** PLA and PVA blending composition.

Code	PLA (wt.%)	PVA (wt.%)
L7V3	70	30
L6V4	60	40
L5V5	50	50

**Table 2 polymers-12-01083-t002:** Thermal data obtained from the second scanning DSC thermograms.

Code	T_g_ (°C)	T_cc_ (°C)	T_m,PVA_ (°C)	T_m,PLA_ (°C)	X_c_ (%)
PLA	62.2	108.8	-	169.0	12.6
PLA7E ^1^	62.0	110.3	-	168.6	12.4
PLA7C ^1^	61.8	109.6	-	168.7	12.3
PVA	57.0	-	150.3	-	9.6
L7V3	61.4	106.0	152.0	167.4	9.3
L6V4	61.3	108.2	153.2	167.2	9.2
L5V5	60.7	102.4	152.1	165.9	9.7

^1^ Injected PLA was immersed in hot water (35 °C) for 7 days.

**Table 3 polymers-12-01083-t003:** Mechanical properties of injected PLA, PVA, and their blends.

Code	Flexural Properties			Tensile Properties		
	Modulus (GPa)	Strength (MPa)	Displacement (mm)	Modulus (GPa)	Strength (MPa)	Elongation at Break (%)
PLA	2.7	100.9	10.1	2.9	68.8	10.1
PLA7 ^1^	2.5	95.7	8.4	3.0	69.5	7.3
PVA	5.2	120.8	9.9	3.0	116.1	45.9
L7V3	2.7	75.9	3.8	2.6	66.1	3.8
L6V4	2.6	73.7	4.1	2.9	67.0	3.9
L5V5	2.7	67.5	3.1	3.1	62.3	3.3

^1^ Injected PLA was immersed in hot water (35 °C) for 7 days.

**Table 4 polymers-12-01083-t004:** The relationship between the mechanical properties and the porosity.

Code	Porosity (%)	Flexural Properties		Tensile Properties		
		Modulus (GPa)	Strength (MPa)	Modulus (GPa)	Strength (MPa)	Elongation at Break (%)
L7V3	0.0	2.7	75.9	2.6	66.0	3.8
	0.5	2.5	73.7	2.8	64.4	3.7
	0.3	2.2	64.5	2.7	66.9	3.7
	0.7	1.9	54.1	2.6	61.9	3.6
	1.6	1.7	51.3	2.6	66.2	3.7
L6V4	0.0	2.6	73.7	2.9	67.0	3.9
	6.8	1.5	41.5	2.9	68.1	3.6
	13.4	1.3	42.0	2.7	60.9	3.4
	19.0	1.4	35.6	2.3	47.2	3.3
	25.0	1.3	29.1	2.2	48.9	3.3
L5V5	0.0	2.7	67.5	3.1	62.3	3.3
	13.8	0.9	28.6	2.0	40.0	2.4
	32.0	0.6	16.1	0.8	10.2	1.9
	42.4	0.5	15.5	0.8	11.4	2.2
	43.4	0.3	12.4	0.6	8.2	2

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
