# Peer review of "Morphology, Thermal and Mechanical Properties of Co-Continuous Porous Structure of PLA/PVA Blends by Phase Separation"

_polymers, 2020, doi:10.3390/polym12051083_

Round 1

Reviewer 1 Report

In this manuscript, the morphology, thermal and mechanical properties of of PLA/PVA blends has been studied. It confirms that porous films of PLA can be obtained by removing the PVA through etching in water, which provides a solvent-free method to obtain scaffold of PLA/PVA blend. This may be of great importance for industrial application. There are, however, many problems which should be considered before publishing.

  1. It is said that the blend is partially miscible in crystalline phase. What does this mean? Does it mean co-crystallization? This is actually not the case.
  2. In the introduction part, it is said that “phase inversion from sea-island to co-continuous morphology depends on interfacial tension, viscosity and its ratio, volume fraction, shear rate and phase dimension”. What does “its ratio” mean? I think it is the same as “volume fraction”. Moreover, it depends actually on the preparation pathway and condition remarkably. As reported in J. Phys. Chem. B 2012, 116, 9832, phase inversion of PLLA/PPC blends from dispersed PLLA phase to co-continuous and to dispersed PPC phase can be realized through adjusting the preparation condition. This should be mentioned.
  3. The statement “co-continuous porous structure was governed by phase separation due to an etching process in hot water” may be not correct. Does the “phase separation” occur during etching? I think that “phase separation” occurs during injection molding; the etching just removes a particular phase and results in the formation of pores.
  4. The calculation of PLA crystallinity (Xc) is correct by using the sum of ΔHm and ΔHcc, since the part contributing ΔHcc contributes also to the ΔHm.
  5. The statement “PVA phase was covered by PLA phase due to low concentration” is also not precise for 50/50 blend.
  6. From the low magnification SEM images shown in Figure 2, phase inversion from sea-island to co-continuous morphology cannot be concluded unequivocally. Enlarged SEM images, also for Figure 8, are needed. Actually, the SEM images shown in Figure 11 do not suggest a co-continuous morphology.
  7. “A degree of crystallinity of PLA of soaked PLA was insignificant ‘change’ (should be ‘changed’) from injected PLA” is hard to be understood. Does it mean “The crystallinity of PLA in soaked PLA/PVA was insignificant changed compared to the injected PLA/PVA”?
  8. There is also problem with the statement “it can be etched by water because it obtained hydroxyl group so phase separation by etching process in water is solvent-free method to obtain scaffold of polymer blend”. If I understand right, it may mean “PVA can be etched by water because it contains hydroxyl group, so the PVA in the phase separated sample can be removed by etching in water, which provides a solvent-free method to obtain scaffold of PLA/PVA blend.
  9. There many English editing problems which should be corrected carefully. Some of them have been listed below only for examples.

  • They were dry mixed at different compositions, which it are presented in Table 1, --> They were dry mixed at different compositions, which are presented in Table 1
  • L7V3 morphology revealed sea-island phase while L6V4 and L5V5 morphology preferred to be co-continuous phase --> L7V3 exhibits a sea-island morphology while L6V4 and L5V5 show co-continuous phase morphologies.
  • L5V5 performed the maximum absorption energy which it was obtained from micro indention --> L5V5 performed the maximum absorption energy as ‘obtained’ (or ‘judged’) from micro indention
  • thermal “induce” phase separation --> thermal “induced” phase separation
  • Scaffold is provided to support during tissue reorganization --> Scaffold provides support during tissue reorganization
  • PVA is widely used many industrial applications --> PVA is widely used in many industrial applications
  • to obtain dumbbell and sheet shape --> to obtain dumbbell and sheet shapes
  • According to low conductive material --> Owing to the low conductivity of PLA/PVA
  • Therefore, PLA phase was more deformation than PVA phase during melt blending --> Therefore, PLA phase was more deformed than PVA phase during melt blending

Reviewer 2 Report

Abstract: “…injected PLA and the blends were performed porous by phase separation in hot water (35 oC) for 0 – 7 days to etch PVA.” Why the injected PLA has the porous? “The maximum etched PVA contents were 0.5, 13.4, 36.1%, respectively” respectively what? In addition, in the text, the values are 0.5%, 13%, and 36%.

“Hence, are utilized and have been widely proposed such as polyvinyl alcohol (PVA), poly(ethylene oxide) (PEO) and gelatin”. The following work

For the co-continuous samples, how to confirm the PVA etched all.  

Figure 2c shows sea-island morphology, and the Figure 2e shows a co-continuous morphology. However, it is difficult to distinguish for the reviewer.

“…order of its value at experimental shear rate of 8 1/s (rotation speed of 500 rpm).” How to calculate the shear rate, please explain?

Some work about the water-soluble polymer blends, PLA and scaffold may be interesting to the authors: Chinese Chemical Letters 2012, 23,351; ACS Sustainable Chemistry & Engineering 2018, 6, 12580; Chinese Chemical Letters 2020, 31, 617; Journal of Renewable Materials, 2020, 8, 89.

Reviewer 3 Report

Comments to the authors:

The manuscript entitled with “Morphology, thermal and mechanical properties of co-continuous porous structure of PLA/PVA blends by phase separation” systematically studied the PLA/PVA blends on their morphologies, thermal and mechanical properties. Etching of PVA from the blends in hot water gave rise to interesting co-continuous structures in the final materials. The results are instructive and important to deeply understand the influencing parameters on the morphology transfer from sea-island to co-continuous morphologies in such blend materials. I recommend acceptance of the manuscript for publication after revising two minor points.

  1. Please remove the editing traces before next submission.

  1. The unit in the title of x-axis in Fig. 16 should be µm but not um.

Round 2

Reviewer 2 Report

In this work, the authors studied morphology, thermal and mechanical properties of PLA/PVA blends. Actually, many papers about the co-continuous morphology of immiscible polymer blends had been well proved for the 50/50, 60/40 and 40/60 blends. In particular, there are also some work about the water-soluble polymer blends, such as ACS Sustainable Chemistry & Engineering 2017, 5, 8334; ACS Sustainable Chemistry & Engineering 2018, 6, 12580. Above works are all used the water-soluble polymer and used water to remove. In addition, the thermal and mechanical properties are not well mentioned in the abstract. The crystallization is not expressed directly the thermal properties. Moreover, the “Flexural and tensile properties considerably decreased with a decrease in the porosity” are also as expected. Accordingly, this work is very traditional one and does not have strong novelty.

Some comments:

In the Title, the co-continuous is used. According to the text, L7V3 is sea-island morphology. So it does not fit the title.

As I said before, “L7V3 exhibits a sea–island morphology, while L6V4 and L5V5 show co-continuous phase morphologies.”; “These polymers exhibited a solitary glass transition temperature, which obeyed the Fox equation.” are not novelty, and they had been proved well in many literature.

In the abstract, “The maximum etched PVA content of L5V5 was 36.1%.”. In the line 377, “…Figure 23 shows a ripped structure of L5V5 with the porosity of 42%.” Why they are different?

The sample preparation temperatures for extruded and injectioned PLA/PVA are respectively 170°C and 180°C. In the line 154: “The PVA phase obtained in extruded L7V3 and L6V4 was covered by the PLA phase owing to low concentration and high viscosity, which is shown in Figure 3.” However, Figure 3 only shows the viscosity at 180°C. The description does not match the data.

The etched PVA content is shown in Figure 5. However, it is not confirm the PVA etched all. For example, the L5V5 in Figure 5 at 7 day is less than 40%. So how to confirm the other 10% PVA etched?

The Figures 18-23 show the relationship between the porosity and mechanical properties. Is makes sense?  

The Figures in this work are too many, some of their can be combined or unnecessary.

Round 3

Reviewer 2 Report

There are still many problems. 

For example, 

In line 173, "The PVA phase obtained in extruded L7V3 and L6V4 was covered by the PLA phase owing to low concentration and high viscosity, which is shown in Figure 3". Based on my prvious commnet, the authors added the viscosity at 170°C. However, according to Figure 3, the viscosities of PLA and PVA are similar. Therefore, the description does not match the data. In addition, the PVA covered by the PLA can not be proved.

The porosity was changed again!!!

The authors saied "In this study, we are using biodegradable polymers with micro/nano-porous structures for their final application in biomaterials and medical devices. In our mention at the introduction, these papers are few reported except for your suggested references (ACS Sustainable Chemistry & Engineering 2017, 5, 8334) and A. Kramschuster and L. S. Turng [ref. 20]." In this work, there is no appliation, and the reviewer thinks that there are also many references, not only above two.

Both the logic and the structure of the manuscript need very further improvement. 

Thus I think this manuscript should be prepared with more time. 
